# Fabrication of Hollow Nanocones Membrane with an Extraordinary Surface Area as CO_2_ Sucker

**DOI:** 10.3390/polym14010183

**Published:** 2022-01-03

**Authors:** Waleed A. El-Said, Jin-Ha Choi, Dina Hajjar, Arwa A. Makki, Jeong-Woo Choi

**Affiliations:** 1Department of Chemistry, College of Science, University of Jeddah, Jeddah 21589, Saudi Arabia; 2Department of Chemical and Biomolecular Engineering, Sogang University, Seoul 04107, Korea; jhchoi@jbnu.ac.kr; 3Department of Biochemistry, College of Science, University of Jeddah, Jeddah 21589, Saudi Arabia; Dhajjar@uj.edu.sa (D.H.); Amaki@uj.edu.sa (A.A.M.)

**Keywords:** hollow nanocones membrane, contact angle, CO_2_ sucker, energy storage, polypyrrole, gas sensor

## Abstract

Recently, more and more attention has been paid to the development of eco-friendly solid sorbents that are cost-effective, noncorrosive, have a high gas capacity, and have low renewable energy for CO_2_ capture. Here, we claimed the fabrication of a three-dimensional (3D) film of hollow nanocones with a large surface area (949.5 m^2^/g), a large contact angle of 136.3°, and high surface energy. The synthetic technique is based on an electrochemical polymerization process followed by a novel and simple strategy for pulling off the formed layers as a membrane. Although the polymer-coated substrates were reported previously, the membrane formation has not been reported elsewhere. The detachable capability of the manufactured layer as a membrane braked the previous boundaries and allows the membrane’s uses in a wide range of applications. This 3D hollow nanocones membrane offer advantages over conventional ones in that they combine a π-electron-rich (aromatic ring), hydrophobicity, a large surface area, multiple amino groups, and a large pore volume. These substantial features are vital for CO_2_ capturing and storage. Furthermore, the hydrophobicity characteristic and application of the formed polymer as a CO_2_ sucker were investigated. These results demonstrated the potential of the synthesized 3D hollow polymer to be used for CO_2_ capturing with a gas capacity of about 68 mg/g and regeneration ability without the need for heat up.

## 1. Introduction

Modern civilization needs extensive use of fossil fuels, hence increases greenhouse gas emissions and CO_2_ gas. This is an urgent challenge worldwide because of its effects on global warming [1,2]. CO_2_ gas is emitted from different sources, including power plants, transportation fuels, and other industries such as petrochemicals, iron, steel, and cement [3,4]. The International Panel on Climate Change (IPCC) reported that the average level of CO_2_ would rise to 570 ppm in 2100, which will cause raising the global average temperature by ≈2 °C [2,5]. Among the different techniques for reducing the CO_2_ level, capturing CO_2_ using solid sorbents is a promising technique [6] because it is eco-friendly, cost-effective, noncorrosive, has high gas capacity, and has lower regeneration energy advantages [7,8]. Several solid adsorbents were used for CO_2_ capture, including activated carbons zeolites [9,10,11,12], pillared clays, mesoporous silica [13], and metal–organic frameworks (MOFs) [14,15,16,17]. However, the rapid decline in the adsorption capacities of these adsorbents limited their use. Furthermore, mesoporous silica/amino organic materials [18,19] were applied to capture CO_2_ based on the interaction between primary/secondary amines and the acidic CO_2_ molecules. This interaction results in the formation of a carbamate ion [20,21]. Thus, the development of new materials with high CO_2_ capture efficiency is urgently needed.

Conducting polymers is characterized by a π-conjugated system with promising physicochemical properties. Thus, they have potential applications including biosensors, batteries, corrosion inhibitors, energy storage, solar cells, light-emitting diodes, and electrochemical supercapacitors [22,23,24,25]. Conducting polypyrrole (PPy) showed many advantages, including (i) excellent electric conductivity, (ii) good chemical and thermal stability, (iii) ease of preparation, (iv) good biocompatibility, (v) low-cost effect, and (iv) eco-friendly nature [26,27,28]. Thus, PPy has been widely used as a cell-based sensor [29,30], as a functional material in energy storage [31,32,33], in drug delivery [34,35], as an electroactive material for sensors [36,37], in actuators [38], in artificial muscles [39], in corrosion inhibitors [40], and in solar cells [41,42].

To control the PPy structure’s morphologies, hard porous membranes were used as a hard template for preparing hollow PPy (hPPy) [43,44]. However, several limitations included the high cost, the risk of damage during the hard templates’ removal process, and the complex preparation process. Thus, more effort has been given to the template-free preparation of micro-/nanostructured hPPy film-modified substrates based on the electrochemical technique during the last decade [45,46,47,48,49,50]. The electrochemical polymerization has various advantages, such as a one-step preparation technique, no need to remove the template after the polymerization, and the morphology/properties of the prepared membrane being easy to control [51,52,53]. Although these studies have successfully reported the fabrication of the hPPy nanocone coated layer, it still needed several materials (i.e., surfactants in strongly alkaline media) and complicated synthesizing steps. A simple, in situ, one-step, and controllable template-free electrochemical technique has not been developed to fabricate the hPPy membrane.

Recently, Ag/hPPy/Ag-nanocomposites-modified Au electrodes were reported as surface-enhanced Raman spectroscopy platform for caspase-3 detection [54]. Here, we reported the fabrication of a three-dimensional (3D) hPPy nanocone membrane for the first time, which showed a significantly larger surface area (949.5 m^2^/g) than any previously reported data for hPPy yet. The developed method is simple, in situ, one-step, and controllable template-free electrochemical technique for fabricating the hPPy membrane without a strong alkaline circumstance. The chemical composition and morphology of the prepared membrane were investigated. This membrane was used for CO_2_ capturing that showed a high affiant and hydrophobic characteristic that avoids moisture adsorption.

## 2. Materials and Methods

### 2.1. Materials

Au-coated glass substrates (50 nm of Au/2 nm of Cr/glass wafers) were purchased from G-mek (Korea). Pyrrole and phosphate buffer saline (PBS) (pH 7.4, 10 mM) were purchased from Sigma-Aldrich (St. Louis, MO, USA). Lithium perchlorate (LiClO_4_) was purchased from Janssen Chimica. All other chemicals were obtained commercially as reagent-grade and used without any further purification.

### 2.2. Fabrication of Hollow PPy Nanocone Membrane

Au-coated glass substrates (20 mm × 10 mm, width × length) were cleaned using acidic piranha solution (3:7, H_2_O_2_:H_2_SO_4_) at 70 °C for 5 min. Then, the substrate was rinsed with deionized water (DIW) and ethyl alcohol and dried under N_2_ gas. Electropolymerization of pyrrole to form hPPy membrane was achieved in 0.1 M of the pyrrole containing 0.1 M of LiClO_4_. The cyclic voltammetry (CV) was applied within a potential range from −0.8 V to +1.2 V at a scan rate of 100 mV/s vs. Ag/AgCl electrode [54]. The active area for the electrochemical polymerization of pyrrole to form hPPy nanocones over the Au electrode was 10 mm × 10 mm. However, substrates with larger sizes were also used for some experiments. Furthermore, different concentrations of pyrrole monomer were used to study the effect of the monomer concentrations on the morphology of the resulting polymer. Here, we have shown the effect of three pyrrole concentrations, 0.001, 0.01, and 0.1 M. The DTG-60 Simultaneous Thermogravimetry/Differential Thermal Analyzer (Shimadzu) was used to study the thermal gravimetric analysis (TGA) under air atmosphere. The X-ray diffraction (XRD) of the hPPy polymer was obtained using X-ray PW 1710 control unit Philips anode material Cu (40 K.V, 30 M.A) Optics (Flex Ltd., Friesland, Netherlands): Automatic divergence slit. Furthermore, the Fourier transform infrared spectroscopy (FTIR) spectrum of the prepared polymer was measured using Nicolet 6700 Thermo Fisher Scientific USA spectrophotometer, using the KBr pellet technique. 

### 2.3. Electrochemical Polymerization 

The hPPy was prepared based on electrochemical polymerization using a potentiostat (CHI-660a, CH Instruments, Austin, TX, USA) controlled by Nova software. The electrochemical measurements were performed using a homemade three-electrode system consisting of a bare Au electrode as a working electrode, a platinum wire as the counter electrode, and an Ag/AgCl reference electrode at a scan rate of 100 mV/s. The morphologies of the hPPy films were studied using field emission scanning electron microscopy (FESEM). The FESEM images were recorded using the ISI DS-130C instrument (Akashi Co., Tokyo, Japan). For better capturing the SEM images of samples, the substrates were fixed on the SEM stage with carbon tapes. Pt films were deposited onto the surface of the substrate at room temperature. The sputtering deposition was performed for 15 s under a constant deposition rate. Then, the substrates were being placed into the FESEM chamber. For the cross-section image, a 45-degree stage was used.

### 2.4. CO_2_ Capture Performance

The CO_2_ capture efficiency of the hPPy membrane was studied using thermogravimetric analysis (TGA) in the presence of pure CO_2_ gas at 50 °C. Typically, the platinum sample pan of the TGA was charged with 10 mg of hPPy and kept the temperature at 50 °C for 30 min under pure N_2_ gas to remove any moisture from the hPPy. The CO_2_ adsorption was performed by switching the gas from N_2_ to CO_2_ (99.9%) for a further 60 min. Then, it was switched back to N_2_ to achieve the desorption process at the same temperature for 60 min. The CO_2_ uptake capacity was determined based on the sample’s weight change during the sorption/desorption processes measured using TGA.

## 3. Results and Discussion

### 3.1. Subsection

Recently, hPPy film-modified substrates were reported in the presence of a soft template in strongly alkaline media based on the electrochemical polymerization process [52,53]. Furthermore, we have reported on the fabrication of hPPy-modified Au electrodes without any linker or template as a surface-enhanced Raman spectroscopy platform for caspase-3 detection [54]. Here, we have prepared hPPy films based on a one-step and easy method using electrochemical polymerization without any surfactants followed by a pull off the film, as shown in Figure 1a. In this schematic diagram, nanocones were growing with the increasing cyclic number of electropolymerization. Figure 1b showed the CV behavior for the pyrrole electropolymerization process for 20 cycles. From the CV data at the beginning of the polymerization process, the CV showed a cathodic peak at about −0.28 V and an anodic peak at +0.0 V. These redox peaks disappeared, and a new cathodic peak at about −0.5 V and anodic peak at about +0.1 V were observed during the polymerization process, combined with increasing background potential. These results mean that progress polymerization of hPPy was correlated to electrical conductivity. Figure 1c–e showed the SEM images of the hPPy film formed after 5 cycles using 0.001 M, 0.01 M, and 0.1 M of pyrrole, respectively. The results demonstrated the fabrication of a layer of mono-laps (nanospheres) with varying sizes diameters based on the monomer concentrations. Increasing the cyclic numbers result in more laps and nanocone structures being formed, as shown in Figure 1f–i. These results confirmed that the nanocones were fabricated based on a lap-over-lap technique that included growing the first stage as hollow nanospheres (first lap). Then, another lap was repeatedly developed over the previous one with each cycle to form nanocones structures.

The morphology of the membrane’s bottom was investigated using SEM images, as shown in Figure 2a–c, which confirmed that the cones are opened from both sides. These results illustrated that this polymer layer did not form as a single domain of hollow nanostructures. Numbers of domains were created in which each one was surrounding with a channel or connection. Furthermore, hollow structures (spheres or cones) were also formed over these channels or connections as shown in Figure 2d–g.

This contrasts with the previous studies for the synthesis of hollow polymer-modified substrates [52,53,54], which assumed that these polymers have a large surface area. However, it is not measurable because it was obtained only as a film on a solid substrate. Here, the formed hPPy polymer film possesses an exciting advantage that concerns its detachable advantage on an easy pull-off technique, as shown in Figure 2h,i and Appendix A. Therefore, the surface area of the resultant polymer membrane was measured based on the N_2_ adsorption/desorption technique (Figure 3a). The Brunauer–Emmett–Teller (BET) method was used to calculate the surface area of the prepared materials. The obtained hollow nanocone polymer membranes show a large surface area of about 949.5 m^2^/g. Table 1 listed the previously reported surface area of hPPy compared to that of the present polymer. This data confirmed that the present polymer has the greatest surface area than the reported surface areas for hPPy [55,56,57,58,59]. The adsorption hysteresis (Figure 3a) exhibits type IV isotherm [60,61,62], which confirmed the presence of mesoporous material. The isotherm exhibits hysteresis loops which are attributed to the presence of mesopores in the obtained materials. This H1 hysteresis loop (IUPAC classification), which implies the presence of porous materials consisting of well-defined cylindrical-like pore channels or agglomerates of approximately uniform spheres [60,61,63].

Furthermore, the bottom of the membrane’s morphology was investigated using the SEM image (Figure 2a–c), which confirmed that the cones are opened on both sides.

The XRD and the FTIR techniques were used to investigate the chemical composition of the developed hPPy membrane (Figure 3b,c), which show the characteristics peaks for the hPPy. The FTIR spectrum showed several bands at the wavenumbers of 1460 and 1550 cm^−1^ (symmetric and asymmetric C–C stretching vibrations), 1300 cm^−1^ (C–N stretching vibration), and 1050 cm^−1^ (bending vibration of the C–H bond). In addition to a broadband in the range from 3000 cm^−1^ to 3500 cm^−1^ (the adsorbed H_2_O and N-H of the pyrrole ring) [64]. X-ray diffraction spectra have shown that pure polypyrrole is amorphous with a broad peak centered at around 2θ = 24.84° [65]. Furthermore, the appearance of peaks at 32.68°, 36.76°, and 38.32° [JCPDS: 30-0751] confirmed the presence of LiClO_4_ inside the hPPy film [66].

The thermal stability of the prepared hPPy membrane was investigated using TGA (Figure 3d). The results demonstrated that the membrane is degraded in two steps. The first stage started from 35 °C to about 249 °C that showed a weight loss of about 10%, which was related to the water loss from hPPy. The second stage ended at 600 °C, which was attributed to the thermal degradation of the hPPY backbone.

To investigate the effects of various conditions on the resultant membrane morphology and its different futures, various counterions, supported substrates, and different types of monomers were used. Oxalic acid, HClO_4_, and sulfuric acid were used as counterions instead of LiClO_4_. The SEM images of the hPPy membrane formed in the presence of different counterions are represented in Figure 4, which demonstrated that the hollow structures were obtained only when LiClO_4_ was used as a counterion. The effect of the chemical composition of the used supporting substrate was investigated using different substrates including indium tin oxide (ITO), Au-coated glass, Au/ITO, and stainless steel (SS), as shown in Figure 5a. The SEM images of the hPPy layer on the different substrates were represented in Figure 5b–e, which demonstrates that the hPPy layer has good adhesion with Au, SS, and Au/ITO substrates. In contrast, it has a low adhesion with ITO substrate. The morphology of the resulting polymer also depends on the substrate used. After optimizing all the preparation conditions, the hPPy membrane was prepared on a large scale of Au/glass substrate (3 cm × 10 cm), as shown in Figure 5f. The results revealed the fabrication of hPPy over a 30 cm^2^ area. Hence, we applied the mass production technique with a recyclable advantage for the used substrate numerous times after removing the membrane.

In addition, electropolymerization was performed using the aniline monomer instead of pyrrole. The SEM images of the resultant polymer were represented in Figure 6a,b. These show a thin layer of the polymer with strong adhesion properties but no cone-like structures. Based on these results for constructing the polymer nanocone membrane, pyrrole is the best monomer in the presence of LiClO_4_ as a counterion with a supporting substrate such as Au-coated glass. Interestingly, the formed hPPy membrane possesses a huge surface charge that results in a repulsion force between the species of the hPPy membranes. Furthermore, this surface charge causes an attractive force between the hPPy membrane and the plastic materials, as shown in Appendix A. Hence, the hPPy membrane could be moved using this kind of attraction force, which opens the door for using this hPPy membrane to develop artificial muscles.

The morphology of the prepared hPPy nanocones is like the lotus flowers. Hence, it was expected that hPPy would show a high hydrophobic characteristic. Thus, the contact angle between water and the hPPy-nanocones-modified Au substrates was studied. Figure 6c–h showed the images of the water contact angles with (i) different-hPPy-layers-modified Au substrate formed in the presence of different counterion ions, (ii) the Au/hPPy/Au-modified Au substrate, and (iii) PANI/Au substrate in comparison with the bare Au substrate. These data indicated that hPPy nanocones/Au formed in the presence of LiClO_4_ have the highest contact angle (136.3°). Thus, it possesses a high hydrophobic characteristic. This hydrophobic characteristic was decreased after Au deposition (60.6°), while the bare Au showed a contact angle of about 73.1°.

### 3.2. CO_2_ Capturing

The CO_2_ adsorption/desorption capacity of the hPPy membrane, as shown in Figure 7, indicated that the CO_2_ adsorption capacity is ≈68 mg/g. This high affinity of the hPPy is related to the significant number of amino groups. It was reported that the CO_2_ molecules could interact with primary and secondary amines to form carbamate, as represented in Equations (1) and (2) [67,68,69,70,71,72].
CO_2_ + 2RNH_2_ → RNHCO_2_^−^ + RNH_3_^+^(1)
CO_2_ + 2R_1_R_2_NH → R_1_R_2_NCO_2_^−^ + RR’NH_2_^+^(2)

On the other hand, the organic heterocyclic molecules that contain N atoms such as pyrrole moiety to interact with CO_2_ through the Lewis acid–Lewis base interactions as well as the hydrogen bonding between the positively charged N atoms of the pyrrole and the negatively charged oxygen atoms of CO_2_ [73,74,75,76,77]. Therefore, the N atoms’ high density within the polymer network increases its adsorbent efficiency toward CO_2_ molecules. Furthermore, the molar ratio of water in the ambient air is typically about 100 times the CO_2_ content. Hence, the development of a selective adsorbent is urgently needed to avoid water co-adsorption [78]. Here, the hydrophobicity characteristic of the hPPy membrane plays a vital role in the CO_2_ adsorption efficiency due to its role to prevent wetting of the membrane pores and thus increase the overall mass transfer coefficient [79]. Figure 7 showed increasing the adsorption of CO_2_ until the pseudoequilibrium was reached after 5 min at 68 mg of CO_2_ for each g of hPPy. The adsorption process was performed for 60 min. Then, the desorption of CO_2_ from the hPPy was studied based on switching the gas flow back into N_2_ at the same temperature. The results showed a linear decrease in the amount of adsorbed CO_2_ until reaching complete desorption within 25 min. This behavior indicated that this system has a completely reversible character. Hence, it showed the possibility of hPPy regeneration without applying heat [77,80]. Furthermore, this regeneration of hPPy could be indicated the existence of weak binding between hPPy and CO_2_. This is one of the advantages during the development of an adsorbent in CO_2_ capture that results in energy-saving regeneration tendency.

## 4. Conclusions

We have developed a new polymer membrane with hollow nanocones morphology in the present work that could be applied in different application fields, including biology, chemistry, and environmental applications. One of the good advantages of this polymer is the easily detachable layer that allows us to use it as a rigid template to fabricate different nanostructures over different materials or as a membrane. Our results demonstrated that we had made a breakthrough in the synthesis of porous conducting polymer membranes. The fabricated polymer showed a large surface area (about 949.5 m^2^/g, the highest yet), a large contact angle of 136.3°, high surface energy, and a pull-off ability as a layer. Furthermore, the fabricated membrane showed high efficiency as a CO_2_ sucker with a gas capacity of about 68 mg/g with a regeneration ability without heat applying. That opens the door for several applications, including biosensors, Li-ion batteries, as hard templates for synthesizing different nanomaterials, drug delivery, membrane, artificial muscle, CO_2_ sucker, etc.

## Figures and Tables

**Figure 1 polymers-14-00183-f001:**
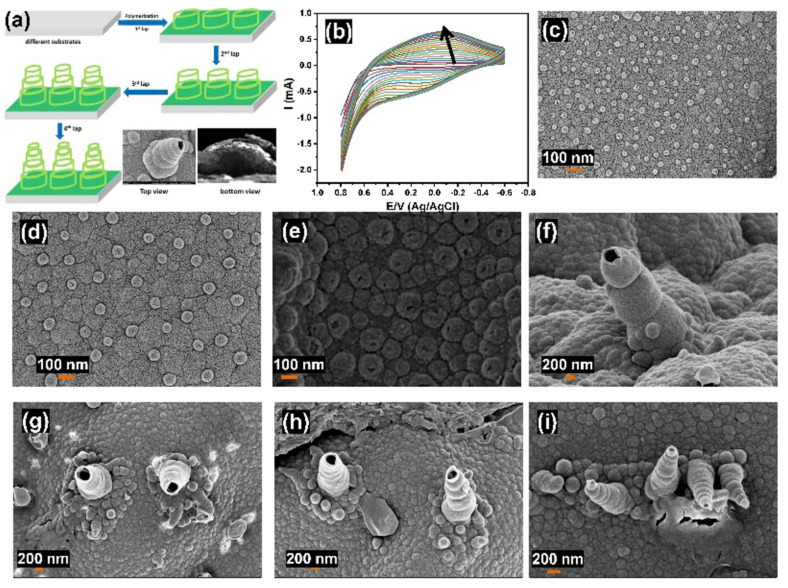
(**a**) Schematic diagram for growing polymer hollow nanocones, (**b**) electrochemical polymerization of PPy, (**c**–**e**) SEM images of polymer nanocones growing based on monomer concentration (0.001 M, 0.01 M and 0.1 M), and (**f**–**i**) effect of the number of cycles on the morphology of polymer nanocones.

**Figure 2 polymers-14-00183-f002:**
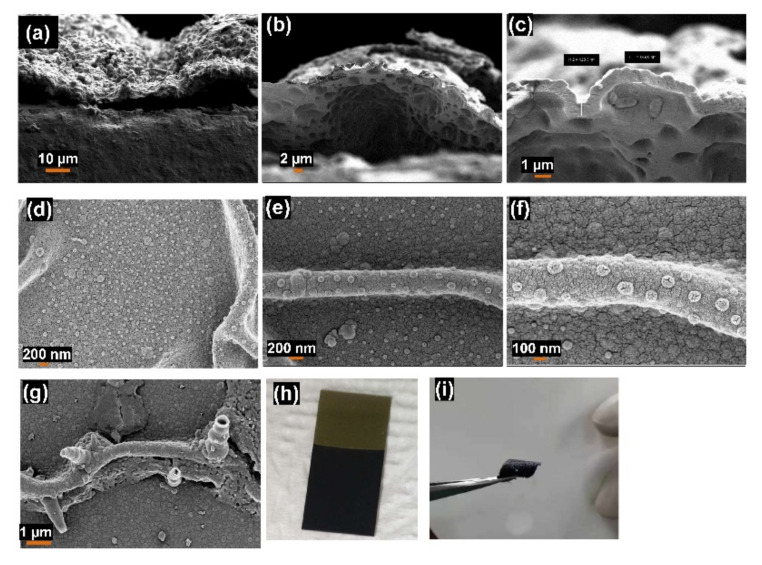
(**a**–**c**) SEM images of a cross-section of the hPPy membrane that showed the morphology of polymer nanocones from the bottom, (**d**–**g**) SEM images of vertical nanocones and horizontal channels (connections), and photography images hPPy nanocones membrane before (**h**) and after (**i**) pull-off from Au substrate.

**Figure 3 polymers-14-00183-f003:**
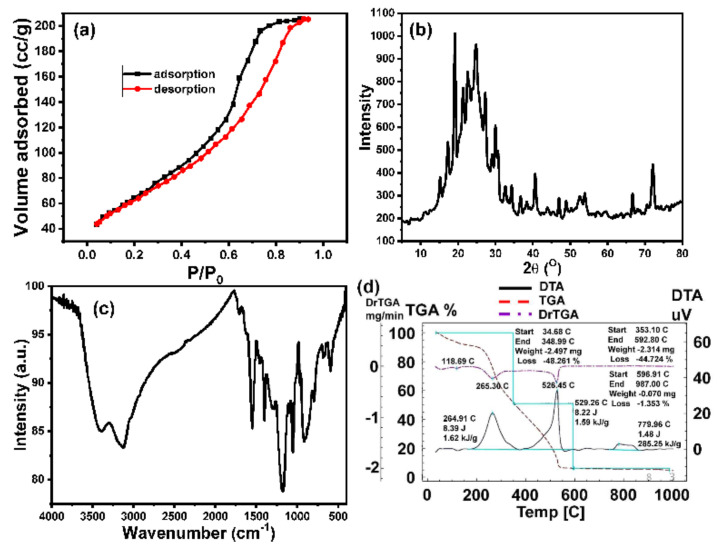
(**a**) N_2_ adsorption/desorption isothermal (surface area) of hPPy nanocones membrane, (**b**) XRD pattern, (**c**) FTIR of hPPy nanocones membrane, and (**d**) TGA of hPPy nanocones membrane.

**Figure 4 polymers-14-00183-f004:**
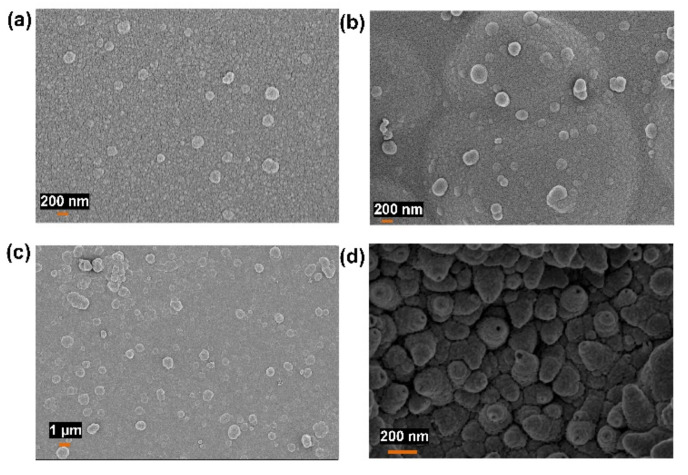
SEM images showed the morphology of hPPy that formed in the presence of (**a**) oxalic acid, (**b**) HClO_4_, (**c**) H_2_SO_4_, and (**d**) LiClO_4_ as counterion ions.

**Figure 5 polymers-14-00183-f005:**
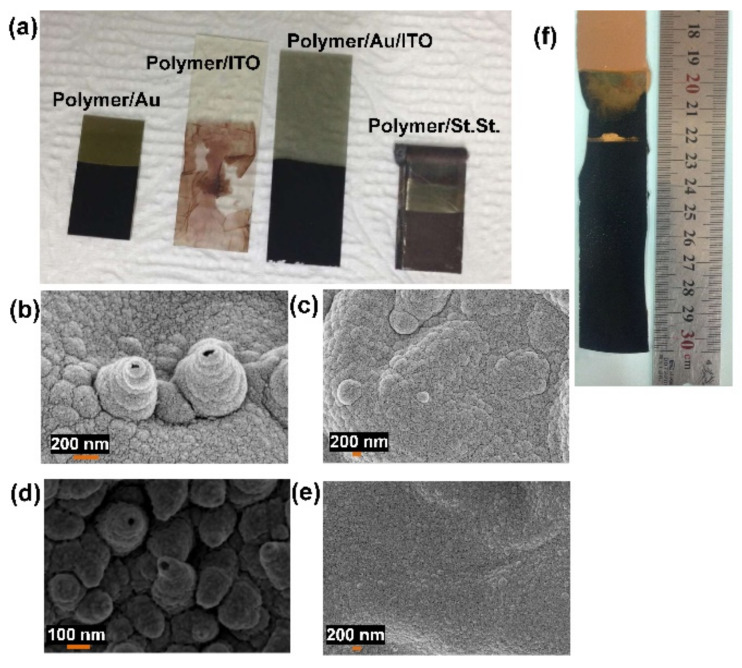
(**a**) Photography images hPPy on different substrates (Au, ITO, Au/ITO, and stainless steel substrates), (**b**–**e**) SEM images of hPPy on Au, ITO, Au/ITO, and stainless steel substrates, respectively, and (**f**) photography image of a large scale of hPPy-nanocones-modified Au substrate.

**Figure 6 polymers-14-00183-f006:**
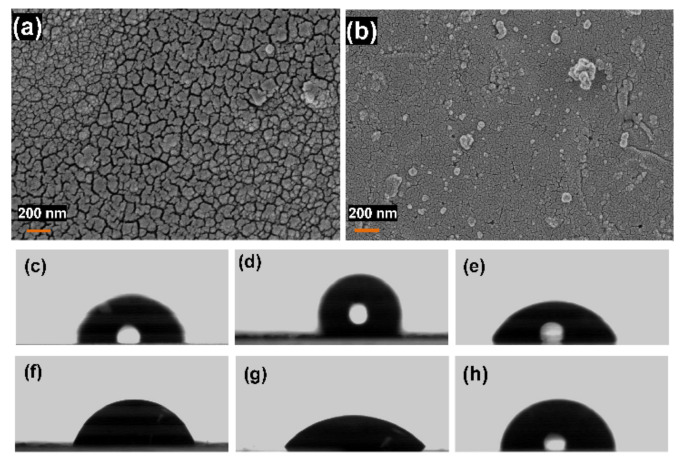
SEM images of (**a**) SEM images of PANI/LiClO_4_, (**b**) SEM images of Au/PANI-modified Au substrate, (**c**) the contact angle of bare Au substrate, (**d**) the contact angle of hPPy-nanocones-modified Au substrate in the presence of LiClO_4_, (**e**) the contact angle of hPPy-nanocones-modified Au substrate in the presence of oxalic acid, (**f**) the contact angle of hPPy-nanocones-modified Au substrate in the existence of HClO_4_, (**g**) the contact angle of hPPy-nanocones-modified Au substrate in the presence of H_2_SO_4_, and (**h**) the contact angle of PANI/Au substrate.

**Figure 7 polymers-14-00183-f007:**
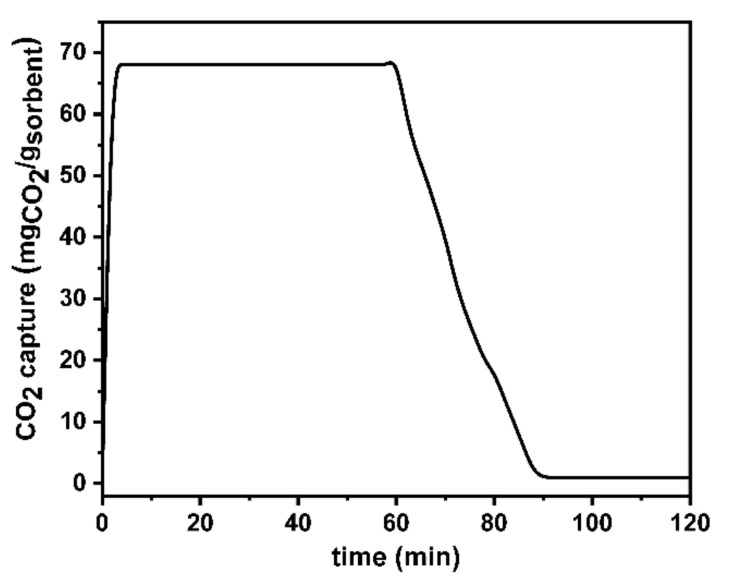
CO_2_ capture by 10 mg of hPPy sample.

**Table 1 polymers-14-00183-t001:** The surface area of reported polypyrrole and its composites.

Material	Surface Area (m^2^/g)	References
PPy	19.2	55
PPy	10.57	56
PPy/cellulose composite	57	57
Nanocellulose PPy membrane	80	58
PPy hydrogel/Au composites	26.2	59
hPPy	949.5	The present work

## Data Availability

The data presented in this study are available on request from the corresponding author.

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
