# Peer review of "Fabrication of Hollow Nanocones Membrane with an Extraordinary Surface Area as CO2 Sucker"

_polymers, 2022, doi:10.3390/polym14010183_

Round 1
Reviewer 1 Report
The manuscript deals with the synthesis and characterization of hollow nanocones membranes with a simple electrochemical process and potentially usable for the capture of carbon dioxide from the surrounding environment.
Specifically, the authors focus on high surface polypyrrole-based membranes systematically studied in terms of thermal and structural properties and their ability to capture CO2.
The research is certainly relevant and the topics covered are worthy of being deepened as the new knowledge can provide a valuable contribution in terms of environmental protection. However, the text is written in poor English which makes difficult to capture the novelty aspects, understand its contents even for competent readers and highlight its real added value compared to the state of the art of knowledge.
Therefore, major revisions are strongly suggested to thoroughly check the entire manuscript, perhaps with the help of a native speaker colleague, with the aim of revising grammatical and syntactic aspects, rephrasing long and incomplete sentences and removing various typos still present in the text.
Reviewer 2 Report
This manuscript needs additional English language editing. There are several grammatical mistakes. Overall the authors have presented a comprehensive paper that expands the chemistry of CO2 absorption. The experimental section is well presented and the conclusions support the experimental observations of this paper. Overall this paper should be published
Reviewer 3 Report
This paper is very interesting because it shows the preparation of a hollow nanocone membrane with a large surface area and its potential as a CO2 adsorbent. In order to improve the quality of the paper, please answer the following questions.
1. The shape of the nanocones on the surface is interesting, but considering the density of the nanocones observed in the SEM image, the effect on the surface area seems to be limited. Many folds are not nanocone, so the effect of the hollow structure, including the areas that are not nanocone, may be significant, as shown in the authors' analysis.
2. For the adsorption of carbon dioxide, please clarify whether the shape of the nanocone is meaningful or whether it can be explained by the increase in surface area alone.
3. Only the open pores contribute to the adsorption of carbon dioxide. Figure 2b-c does not clearly show whether the pores are closed or open. Please discuss the morphology of the pores in terms of porosity, pore size, and surface area.
In addition, the following technical fixes are required.
4. In the PDF file, Figure 5 and Figure 6 are overlapped.
5. There is no scale bar in the SEM images of Figures 1, 4, and 5.
I don't know about Figure 6 because I can't see all of it, but Figure 6 may need a scale bar as well.
6. The 2 of CO2 in the abstract and keywords should be subscripted.
7. The 2 of m2/g in the abstract should be superscripted.
8. The numbering of the references is not in the order of the citations; 36 is used before 22.
9. In L161, "Figures 2h&2i and S1" should be "Figures 2h&2i and Video S1".
10. In L208, "Figures S2-4" should be "Videos S2-4".
Reviewer 4 Report
All the Figures should be redrawn, especially the SEM images, and several experiments need to be done.

Reviewer 5 Report
Nowadays, the environmental problem resulted from the massive emission of CO2 is more and more serious. In this manuscript, authors reported a new polymer membrane with the 3D hollow nanocones morphology, which showed a large surface area, a large contact angle, high surface energy, and pull-off ability. The CO2 capture efficiency was high as well. However, the writing of this manuscript must be improved because there are a lot of grammar mistakes and typos. In addition, it is better to add more detailed explanations.
(1) Please notice the subscripts in the Abstract: CO2 -> CO2
(2) Please add the full name before using the abbreviation, such as 3D in the Abstract.
(3) Please unify the units (i.e. 949.5m2/g, 949.5m2/g, and 949.5m2/gm).
(4) Line 18: has breaks or has broken?
(5) Line 97 & 99: please show the full names of XRD and FTIR.
(6) Line 92 & 106: please unify the unit (100 mV/s or 100 mV/S).
(7) Line 122, hPPy film -> hPPy films.
(8) A lot of SEM results are discussed in the manuscript, please add more details of SEM experiments in Section 2 (Materials and Methods).
(9) Line 131: the PPy film was formed after 5 cycles by using different monomer concentrations. What concentrations did authors use and why did authors select these concentrations? Please add more details in the experimental section.
(10) In Fig. 1f-1i, the nanocones were fabricated based on lap-over-lap. Could authors explain why this happened? Why was the diameter of the top loop smaller than the bottom one's? Please add more theoretical explanations of the nanocone fabrication.
(11) Please modify the SEM images in all the figures the same as the SEM images in Fig. 2. Please add scale bars in the SEM images and delete the bottom annotations of SEM images like SEM images in Fig. 2.
(12) Line 150 & 151: there are two "was" in the sentence.
(13) Please add the scale bars in Fig. 2h & 2i to let readers know the size of the samples.
(14) TGA (Fig. 3d) is not clear in the manuscript. Could authors try to make it clearer by using other software (i.e. Origin) to remake a figure?
(15) XRD and FTIR are important results to show the properties of the polymer. Please explain Fig. 3b & 3c more in detail rather than only one sentence.
(16) Line 231: this -> This.
(17) Could you explain why the CO2 capture decreased after 60 mins? Why did the CO2 capture drop down to 0 in 25 mins? In Fig. 7, the decrease of CO2 capture from 60 mins to 85 mins seems to be linear, so was the decrease at a constant speed? Why?
Round 2
Reviewer 1 Report
The revised manuscript is acceptable for publication.
Author Response
Many thanks for your comments. We are really excited that you found that my manuscript is suitable for Polymers.
Reviewer 3 Report
The manuscript has been improved significantly.
Author Response

(The authors gave the same response as above.)
